# Transposable Element Landscape in the Monotypic Species *Barthea barthei* (Hance) Krass (Melastomataceae) and Its Role in Ecological Adaptation

**DOI:** 10.3390/biom15030346

**Published:** 2025-02-27

**Authors:** Wei Wu, Yuan Zeng, Zecheng Huang, Huiting Peng, Zhanghai Sun, Bin Xu

**Affiliations:** 1College of Horticulture and Landscape Architecture, Zhongkai University of Agriculture and Engineering, Guangzhou 510225, China; zengyuan025@gmail.com (Y.Z.); luckzechengh@gmail.com (Z.H.); penghuiting.001@gmail.com (H.P.); sunzhanghai.17@gmail.com (Z.S.); 2Guangdong Provincial Key Laboratory of Silviculture, Protection and Utilization, Guangdong Academy of Forestry, Guangzhou 510520, China

**Keywords:** transposable elements, ecological adaptation, genome evolution, *Barthea barthei*, TE insertion polymorphisms

## Abstract

Transposable elements (TEs) are crucial for genome evolution and ecological adaptation, but their dynamics in non-model plants are poorly understood. Using genomic, transcriptomic, and population genomic approaches, we analyzed the TE landscape of *Barthea barthei* (Melastomataceae), a species distributed across tropical and subtropical southern China. We identified 64,866 TE copies (16.76% of a 235 Mb genome), dominated by Ty3/Gypsy retrotransposons (8.82%) and DNA/Mutator elements (2.7%). A genome-wide analysis revealed 13 TE islands enriched in genes related to photosynthesis, tryptophan metabolism, and stress response. We identified 3859 high-confidence TE insertion polymorphisms (TIPs), including 29 fixed insertions between red and white flower ecotypes, affecting genes involved in cell wall modification, stress response, and secondary metabolism. A transcriptome analysis of the flower buds identified 343 differentially expressed TEs between the ecotypes, 30 of which were near or within differentially expressed genes. The non-random distribution (primarily within 5 kb of genes) and association with adaptive traits suggest a significant role in *B. barthei*’s successful colonization of diverse habitats. Our findings provide insights into how TEs contribute to plant genome evolution and ecological adaptation in tropical forests, particularly through their influence on regulatory networks governing stress response and development.

## 1. Introduction

Transposable elements (TEs) constitute a substantial proportion of most plant genomes. Active TEs mobilize within the host genome through either ‘copy and paste’ (Retrotransposons, Class I) or ‘cut and paste’ (DNA transposons, Class II) mechanisms [1]. Retrotransposons utilize an RNA intermediate, while DNA transposons employ a DNA intermediate. Both classes are further categorized as autonomous (self-mobilizing) or non-autonomous (dependent on other TEs). A hierarchical classification system organizes TEs into orders, superfamilies, families, and subfamilies based on their sequence structure and phylogenetic relationships [2]. In plant genomes, the long terminal repeat (LTR) retrotransposon order, comprising the Copia (RLC) and Gypsy (RLG) superfamilies, are typically the most abundant and range in size from a few hundred base pairs to 25 kilobases (kb) [3]. Other retrotransposon orders include non-LTR elements such as long interspersed sequences (LINEs), short interspersed sequences (SINEs), dictyostelium intermediate repeat sequences (DIRS), and Penelope-like elements (PLEs) [3]. Class II elements are classified into the terminal inverted repeat (TIR) and Helitron orders. TIR transposons are further divided into superfamilies based on target site duplications (TSDs) and terminal inverted sequences, including Tc1–Mariner (DTT), hAT (DTA), Mutator (DTM), P (DTP), PIF–Harbinger (DTH), and CACTA (DTC) [3]. Miniature inverted-repeat transposable elements (MITEs) are non-autonomous DNA transposons characterized by a short length (<500 bp) and a high copy number. While common in plant genomes (e.g., *Stowaway* and *Tourist* families in rice [4]), MITEs share structural features with diverse TEs; thus, their designation lacks taxonomic significance [3]. Helitrons, another Class II order, utilize a unique rolling-circle mechanism for transposition. They are defined by a 5′-TC and 3′-CTRR motif, a short hairpin structure 15–20 bp upstream of the 3′ end, and the frequent duplication of passenger genes [3].

Once dismissed as ‘junk DNA’ or ’parasites’, TEs are now recognized as key drivers of genome architecture and evolution [5,6]. Genome size variation across plant species is largely attributable to TE content, ranging from <3% in the small genome of *Utricularia gibba* (77 Mb) [7] to >70% in the massive *Picea abies* genome (19.6 Gb) [8]. While a positive correlation between genome size and TE diversity was initially expected, the relationship only holds true for genomes smaller than 500 Mb [9]. For example, despite a low TE family diversity, the *P. abies* genome is dominated by a few LTR families with high copy numbers that are estimated to be 5–60 million years old (MYA). Conversely, medium-sized genomes with high TE diversity often exhibit rapid turnover, with most TEs being <5 MYA [8,10]. The factors driving this variability in TE diversity and the success of specific TE superfamilies/families in certain lineages remain poorly understood [11]. However, the growing availability of plant genome sequences offers the potential to characterize TE landscapes across broader phylogenetic scales and elucidate the environmental and genetic mechanisms underlying the observed variations.

Transposable elements are a significant source of genetic variation, contributing to phenotypic innovation and adaptation through mechanisms such as domestication, exaptation, host gene regulation, retrogene formation, and enhanced genomic plasticity [12,13,14]. While TEs are typically silenced via DNA methylation, histone modification, small RNA-based silencing, and chromatin modifications [15], they can be reactivated by stress or developmental cues [16,17]. Diverse TEs have been shown to regulate various agronomically and ecologically relevant traits, including flower color in morning glories [18], fruit coloration in apples [19], sex determination in melons [20], drought tolerance in maize seedlings [21], and photoperiod sensitivity in maize [22]. Although these TE-mediated trait variations are primarily documented in crops and model species, their impact in natural populations, where diverse abiotic and biotic factors influence TE expression, may be underestimated. The growing availability of non-model plant genomes promises to reveal further TE diversity.

*Barthea barthei*, an evergreen shrub and the sole member of the monotypic genus *Barthea* (Melastomataceae), is endemic to southern mainland China as well as Taiwan Province [23]. While two varieties were previously distinguished based on capsule wing width, a recent population genetic analysis does not support this classification [24]. Unlike the uniformly red flowers of *Melastoma* species, *B. barthei* exhibits flower color polymorphisms, with red and white flowers observed in natural populations spanning altitudes of 400–2500 m (Figure 1A). Red-flowered populations typically inhabit open forest areas or mountaintops, while white-flowered populations are usually found in the shaded understory. These two groups are designated the ‘red ecotype’ and ‘white ecotype’, reflecting their adaptation to differing light intensities. While the role of TEs in genome evolution is well-established, their contribution to adaptive traits, particularly in non-model plant species, remains poorly understood. In particular, the link between TE variation and ecologically relevant traits, such as flower color polymorphisms, requires further investigation. *B. barthei*, with its distinct red and white flower ecotypes adapted to different light environments, provides an excellent system to explore this connection. Therefore, this study aims to address the following key questions: (1) What is the TE landscape of *B. barthei*, including TE abundance, types, and genomic distribution? (2) Are there differences in TE content or expression between the red and white ecotypes? (3) Do specific TE insertions correlate with differential gene expression and potentially contribute to the observed flower color polymorphism? To answer these questions, we characterize the *B. barthei* TE landscape using a published genome assembly. Integrating RNA-seq and population genome resequencing data from the two ecotypes, we investigate the potential role of TEs in adaptation to contrasting light environments (Appendix A).

## 2. Materials and Methods

### 2.1. Plant Material and Sequencing

Flower buds from two *B. barthei* ecotypes were collected from natural populations. One ecotype with white petals was sampled from Erhuangzhang Nature Reserve, Yangcun County, Guangdong, China (YC population, 111°25′22.59″ E, 21°53′14.37″ N, alt. 611 m) and the other red ecotype from Lianhua Mt. Huidong county, Guangdong, China (HD population, 115°13′56.10″ E, 23°3′42.43″ N, alt. 1280 m). Flower buds from each ecotype for three individual plants were sampled, immediately frozen in liquid nitrogen in the field, and taken back for total RNA isolation using the RNAprep Pure Plant Kit (TIANGEN Biotech Co., Ltd., Beijing, China). The quality of total RNA was determined using Agilent 2100 Bioanalyzer (Agilent Technologies, Palo Alto, CA, USA). The qualified RNAs were subject to paired-end library construction using Illumina TruSeq RNA Sample Preparation Kit (Illumina, San Diego, CA, USA), and sequenced on the Illumina Hiseq2500 platform using 150 bp × 2 paired-end reads. Additionally, we utilized existing genome resequencing data from nine HD and eleven YC individuals generated in a previous study [25].

### 2.2. Transposable Element Annotation of the Genome B. barthei and Phylogenetic Analysis

The chromosome-level *B. barthei* genome assembly, generated using PacBio Sequel II platform and high-throughput chromatin conformation capture mapping [25], was used for TE annotation. A non-redundant TE library was curated and the genome annotated using the Extensive de novo TE Annotator (EDTA) package [26]. Unknown superfamily LTR and non-LTR retrotransposons were classified using DeepTE with a default probability threshold of 0.6 [27]. Retroelements were searched against NCBI’s conserved domain database (CDD, http://www.ncbi.nlm.nih.gov/Structure/cdd/cdd.shtml (accessed on 22 January 2024)) [28] using RPSBLASTN v.2.11.0+. Elements matching Pfam00078 (RVT_1, Ty3/Gypsy) or Pfam07727 (RVT_2, Ty1/Copia) with a minimum of E-value of 0.001 were retained. For family-level classification of these filtered retroelements, one RT domain protein sequence from each family in the REXdb (Viridiplante v3.0) [29] was randomly selected. Open reading frames (ORFs) within the corresponding nucleotide sequences were identified using Open Reading Frame Finder program (RRID:SCR_016643, https://www.ncbi.nlm.nih.gov/orffinder (accessed on 23 January 2024)), and amino acid sequences > 50 residues were aligned using MAFFT v7.407 with default settings [30]. ModelTest-NG v0.1.7 [31] was used to determine the appropriate amino acid substitution model before maximum likelihood phylogeny inference using RAxML v8.2.12 [32] with 100 nonparametric bootstrap replicates. Custom scripts are available on GitHub (https://github.com/altingia/Barthea_TE_manuscript accessed on 23 January 2023).

### 2.3. Insertion Time Calculation of Intact LTRs and History of TE Proliferation Inference

The insertion time of intact LTR retrotransposons was estimated by calculating the divergence between their initially identical long terminal repeats. LTRpred v1.1.3 [33], optimized for intact LTR retrotransposon detection, was used to calculate insertion ages based on a mutation rate of 1.3E-8 substitutions per site per year [34]. Kimura genetic distances between individual TE family copies and their consensus sequences were retrieved from RepeatMasker-v4.1.6 output using the ‘parseRM.pl’ script. These distances were used to summarize and visualize the accumulated coverage of TE classes and subclasses across different age ranges, providing insights into TE proliferation history. Custom R scripts used for this analysis are available on GitHub (https://github.com/altingia/Barthea_TE_manuscript/blob/main/03.TE_distributions/3.2.TE_islands/500k_sliding/Copia.windows.coverage.bed (accessed on 23 January 2024)).

### 2.4. Relationship Between Transposable Elements and Adjacent Genes and Transposable Element Islands in the Host Genome

To identify TEs located near gene promoters, a *Barthea*-specific transcription start site (TSS) model was trained using TSSFinder v1.0.0 [35] based on the Arabidopsis model. TSSs with a TATA-box motif in the core promoter region were predicted, and TEs located ≤ 1000 bp upstream of a TSS were considered promoter insertions. A custom pipeline was developed to determine the relationship between each TE and its nearest gene, classifying them as ‘upstream (promoter)’, ‘5-overlap’, ‘5-UTR’, ‘CDS’, ‘intron’, ‘3-UTR’, ’3-overlap’, or ‘downstream’ (details seen in Appendix A). For nested TEs, spanned regions were determined by integrating TE coordinates into the gene feature coordinates array. This allowed precise mapping of TE and gene feature relationships (https://github.com/altingia/Barthea_TE_manuscript/tree/main/04.TE_gene_relationships accessed on 22 January 2024).

The frequency of each category and their associated distances/overlap lengths were summarized for different TE classes. TE abundance was analyzed using 500 kb sliding windows (100 kb step size) along each linkage group. Windows with >50% TE content were defined as TE islands, and consecutive islands were merged. Gene ontology (GO) terms and KEGG pathways were assigned to genes within TE islands (foreground) and the entire genome (background) using KOBAS v3.03 [36] with an E-value threshold of 1 × 10−3. Enrichment analysis was performed, and significance was assessed using the Benjamini–Hochberg (BH) test and False Discovery Rate (FDR) correction [37].

### 2.5. Transcriptome Profiles for Transposable Elements and Genes Between White and Red Ecotypes

RNA-seq reads from flower buds (three biological replicates per ecotype) were trimmed and filtered using Trimmomatic v0.39 [38] with default parameters (ILLUMINACLIP: TruSeq3-PE.fa:2:30:10, LEADING:3, TRAILING:3, SLIDINGWINDOW:4:15, MINLEN:36, TOPHRED33). Trimmed reads were mapped to the reference genome using STAR 2.7.9 [39] with parameters settings ’—outFilterMultimapNmax 100—winAnchorMultimapNmax 200—outSAMtype BAM SortedByCoordinate’, allowing for multiple mapped reads, as is recommended for TE analyses. Both TE and gene expression levels were normalized to counts per million (CPM). Differential expression analysis between ecotypes was performed using TEtranscripts v2.0.3 [40] with the following parameters: ’-stranded no-mode multi −*p* 0.05 −i 10’. Only genes or TEs with reads spanning their entire length were considered transcribed.

### 2.6. Transposable Element Insertion Polymorphisms Among B. barthei Populations

TE insertion polymorphisms (TIPs) were identified using SPLITEREADER beta-1.2 [41] and TEPID v0.10 [42], following established recommendations for specificity and sensitivity [43]. We used the programs to identify non-reference insertion sites and reference absence variants. SPLITREADER identifies potential non-reference insertions, which are then merged and filtered by TE family, retaining sites with ≥3 supporting reads in at least one individual. Both pipelines examine negative coverage patterns in flanking regions (100 bp upstream and downstream) to distinguish presence and absence variants. True non-reference insertions exhibit reduced coverage, while true absence variants show coverage drops at their edges. Low-coverage regions lacking sufficient evidence for either presence (positive coverage) or absence (negative coverage) are marked as NA. Single nucleotide polymorphisms (SNPs) were called using the GATK pipeline (The Broad Institute, Cambridge, Massachusetts, USA) following best practices [44,45]. Reads were aligned with BWA-MEM, duplicates were marked with Picard, and base quality scores were recalibrated with GATK’s BaseRecalibrator. GATK HaplotypeCaller (The Broad Institute, Cambridge, MA, USA)was used for variant calling, and SNPs were filtered (QD < 2.0, FS > 60.0, MQRankSum < −12.5, ReadPosRankSum < −8.0, and QUAL < 30.0). Principal component analysis (PCA) of TIPs was performed using the ‘prcomp’ function from the stats package v3.2.3 in R v3.4.4 [46], retaining the first two eigenvectors. Linkage disequilibrium (LD) for each TIP was analyzed using PLINK v1.9 [47]. The LD coefficient (r^2^) was calculated between each TIP and 300 flanking SNPs (upstream and downstream), as well as pairwise r^2^ values among all 600 SNPs. The proportion of TIP–SNP pairs with high LD (r^2^ > 0.4) was compared to the proportion of high LD SNP–SNP pairs. Similar patterns were observed using a less stringent threshold of r^2^ > 0.2. To account for regional LD variation, TE variants were categorized as ‘low’, ‘medium’, or ‘high’ LD based on their ranked of TIP–SNP r^2^ values relative to the median ranked r^2^ of SNP–SNP pairs in the same region.

## 3. Results

### 3.1. Composition and Proliferation History of Transposable Elements in the B. barthei Genome

Using EDTA v2.0.1, we curated a B. barthei-specific, non-redundant library of 1081 consensus TE sequences. The RepeatMasker analysis revealed a total of 64,866 TE copies, comprising 16.76% of the genome (40% divergence threshold, and 19,316 copies of other repeat elements (low-complexity regions, microsatellites), comprising 2.1%) (Table 1). DeepTE reclassified 5665 of the 5923 initially unclassified LTRs, assigning 3768 to the Ty3/Gypsy and 1897 to the Ty1/Copia superfamilies (Table 1). The unclassified non-LTRs were categorized as 224 PLEs, 95 LINEs, 38 SINE/tRNAs, and 1 DIR. In total, we identified 24,296 Ty3/Gypsy (8.82%) and 9602 Ty1/Copia (2.47%) copies (11.29% total), non-LTRs including 588 LINEs/L1, 8 LINEs/I, and 278 Penelope copies, and 220 tRNA copies (Table 1).

The Class II DNA transposons constituted 5.28% of the genome (3.44% TIRs, 1.63% Helitrons, and 0.21% MITEs) (Table 1). Among the TIRs, Mutator was most abundant (13,868 copies, 2.7%), followed by CACTA (2820 copies, 0.49%), PIF-Harbinger (747 copies, 0.12%), hAT (516 copies, 0.09%), and Tc1-Mariner (127 copies, 0.04%). Of the 64,864 TEs, 1108 (2.1%) were intact, including 120 LTR/Copia, 220 LTR/Gypsy, 1 nonLTR/PLE, 426 TIR/DTM, 118 TIR/DTC, 81 TIR/DTA, 38 TIR/DTH, 15 TIR/DTT, and 89 Helitron. The remaining 14.66% consisted of truncated or fragmented TEs (Table 1).

The TE age distribution peaked sharply at zero, indicating recent proliferation, followed by a steep decline and a broader distribution from 3 to 10 million years ago (MYA) (Figure 1B). The superfamily proliferation, in decreasing order, was as follows: LTR/Gypsy, DNA/Helitron, DNA/DTM, LTR/Copia, DNA/DTC, and DNA/DTH, with most elements < 15 MYA. Of the 782 LTRs with intact flanking long terminal repeats (188 Ty1/Copia, 381 Ty3/Gypsy, and 213 Unclassified), the insertions occurred between 0.001 and 1.064 MYA and were concentrated within the last 1.00 MYA (Figure 1C). The Copia and Gypsy insertion times largely overlapped, peaking < 0.5 MYA, while Unclassified LTRs had an older peak (Figure 1C). These recent, intact LTRs may still be mobile and functional.

### 3.2. Retrotransposon Classification Based on RT Domains

The phylogenetic analysis of the RT domains identified 342 Ty1/Copia (RVT_2) and 214 Ty3/Gypsy (RVT_1) domains. Seventeen Ty1/Copia and twelve Ty3/Gypsy family representatives were used to construct a maximum likelihood phylogenetic tree (PROTGAMMAJTT model, rapid bootstrap). Nine distinct Ty1/Copia lineages were identified, including the abundant *Osser* (74 copies), *Bianca* (73 copies), and *Angela* (47 copies), and less prevalent families such as *Ale* (20 copies), *Sire* (10 copies), *Tork* (12 copies), and *Tar* (12 copies) (Figure 1D). Among the Ty3/Gypsy elements, 108 remained unclassified, while the rest were assigned to *Athila* (36 copies), *Renia* (28 copies), *CRM* (18 copies), *Tekay* (16 copies), *Galadriel* (3 copies), and *Chlamyvir* (1 copy) (Figure 1E).

### 3.3. Identification and Characterization of Transposable Element Islands in Gene Spaces

The TEs were unevenly distributed across the 20 linkage groups. Thirteen TE islands, ranging from 20.9 kb to 1.4 Mb and encompassing 11.7% of the total TE content, were identified (Figure 1E, Appendix A, Appendix A). The largest TE island, located on chromosome 6, spanned 1.4 Mb. These islands contained 1037 protein-encoding genes, exhibiting a significantly lower gene density than the TE-poor regions (Fisher’s exact test, *p* < 2.2 × 10^−16^). However, these regions were enriched in genes involved in photosynthesis (ath00195, FDR = 2.2 × 10^−9^, BH test) and tryptophan metabolism (ath00380, FDR = 0.04, BH test). Twenty-nine GO terms, particularly those related to abiotic stress (e.g., ‘response to hydroxyurea’, ‘response to aluminum ion’, and ‘response to UV-B’), were also significantly enriched in these TE islands (Appendix A). This suggests that *B. barthei* genetic evolution has been shaped by adaptation to the acidic, aluminum-rich soils and variable light conditions of South China.

### 3.4. Genome-Wide Survey of Transposable Element Insertion Preferences in the B. barthei Genome

A genome-wide survey mapped TE insertion sites in *B. barthei* using a custom pipeline (https://github.com/altingia/Barthea_TE_manuscript/tree/main/04.TE_gene_relationships (accessed on 23 January 2024)) (Appendix A, Appendix A). Of the 64,864 identified TE copies, 63,665 were mapped relative to neighboring genes, and 1199 were found on unannotated scaffolds (Table 2). The TEs preferentially inserted upstream (45.8%; 29,144 copies) compared to downstream (41.7%; 26,562 copies). The LTR retrotransposons were frequently located >5 kb from neighboring genes (Appendix A), while the DNA transposons (TIRs and Helitrons) were predominantly within 5 kb of genes (Appendix A). A total of 17,991 upstream and 14,845 downstream TEs were within 5 kb of neighboring genes. Additionally, 577 TE copies were identified within gene promoters, affecting 431 genes (Table 2). Approximately 26.1% of *B. barthei* genes had at least one TE copy within 1 kb, lower than Arabidopsis (36%) or maize (86%) [48,49]. The most common TE types near genes were LTR/Gypsy, DNA/TIRs, LTR/Copia, DNA/Helitron, and nLTR (Table 2). Only a small proportion of the genes with adjacent TEs showed a significant GO term enrichment, including ‘ADP binding (GO0043531)’ and ‘response to ethylene (GO0009723)’ (Appendix A). Additionally, 760 TE copies overlapped with gene 5′ ends and 630 with 3′ ends.

A substantial number of TEs (6569 copies, 10.3%) were nested within host genes (intronic, coding sequence (CDS), or untranslated region (UTR)) (Table 2). Single-intron nested TEs were the most prevalent (5454 copies in 2560 genes). *Barthea36295* (homologous to *Vitis vinifera* LUTEIN DEFICIENT 5, a cytochrome P450 member [50]) contained the most nested TEs (37 copies, 23 in its third intron) (Appendix A). The genes with single-intron nested TEs were significantly enriched for ‘cytosol’ (GO:0005829, FDR = 3.3 × 10^−6^) and ‘cytoplasm’ (GO:0005737, FDR = 3.2 × 10^−4^) (Appendix A).

Exon-nested TEs (635 copies in 227 genes) were predominantly in UTRs (190 in 5′ UTR, 370 in 3′ UTR) rather than CDS regions (75 copies in 43 genes). *Barthea44534* (homologous to beta-glucosidase 12-like [51]) contained the most UTR-nested TEs (22 copies) (Appendix A). *Barthea28554* (homologous to the RNA polymerase beta subunit) contained the most DNA/DTM copies (Appendix A). While eight genes lacked significant hits, 35 others resembled conserved proteins (ribonuclease, reverse transcriptase, plant disease resistance polyprotein, heat shock protein, etc.) (Appendix A). The genes with exon-nested TEs were associated with cellular and stress responses, including protein folding chaperone activity and response to temperature stimulus (Appendix A). Additionally, 480 TE copies spanning multiple introns/exons were found in 618 genes, and these were linked to developmental processes and epigenetic modification (Appendix A).

### 3.5. Expression Profiles of Genes and Transposable Elements During Flower Bud Development Between Contrasting Ecotypes

The RNA-sequencing sample quality was validated by hierarchical clustering, demonstrating a high reproducibility within the ecotypes (Appendix A). The transcriptome analysis identified 40,538 expressed genes and 4495 transcribed TE copies (Figure 2, Appendix A). The differentiation expression analysis (baseMean > 10, |log2FoldChange| ≥ 2, and Padj < 0.05) identified 2544 differentially expressed genes (DEGs), comprising 1357 up-regulated and 1187 down-regulated genes between the two ecotypes (Appendix A).

The KEGG pathway enrichment analysis of the DEGs revealed three significant pathways (Appendix A). ‘Photosynthesis-antenna proteins’ (KO00196, FDR = 3.23 × 10^−8^, BH test) included 20 DEGs (12 Arabidopsis homologs, including LHCA1~LHCA6 and LHCB1~LHCB6) (Appendix A), which are crucial for energy equilibrium under variable light [52]. ’Phenylpropanoid biosynthesis’ (KO00940, FDR = 1.50 × 10^−7^, BH test) comprised 51 DEGs (30 Arabidopsis homologs) (Appendix A), which are involved in lignin and flavonoid synthesis and the light-induced stress response [53]. ‘Cutin, suberine, and wax biosynthesis’ (KO00073, FDR = 2.09 × 10^−4^, BH test) contained 15 DEGs (*Arabidopsis* homologs, including HXXXD-type acyl-transferase family protein, FAR2, CYP86A1, CER1, CYP704B1, and CYP86B1) (Appendix A), which are involved in protective cuticular wax production against environmental stressors [54,55].

Of the transcribed TEs, retrotransposons were abundant (1457 transcripts): 1059 LTR/Gypsy, 206 LTR/Copia, 148 LTR/unknown, and 44 LINEs/unknown. These were frequently near genes or within single introns, with LTR retroelements averaging 5738.5 bp from adjacent genes (Table 3 and Appendix A). DNA transposon transcripts (1938, 43.1% of expressed TEs) were also detected, including 403 DNA/DTC, 626 DNA/DTM, 48 DNA/DTH, 45 DNA/DTA, 27 DNA/DTT, 140 MITEs, and 649 Helitrons. Of these, 855 (19.0%) were within intron/UTR/CDS regions, and the rest averaged 1881 bp from genes (Table 3 and Appendix A). The consistent detection of DNA transposon transcripts across the samples suggests they are not mere DNA contaminants, as previously suggested in other RNA-seq studies [56,57]. Their distribution (42.8% nested, 57.2% within 2000 bp of host genes) indicates probable passive co-transcription. Furthermore, 1083 unclassified TE transcripts were identified, with 391 nested and 605 averaging 7995.8 bp from genes (Table 3).

The differential expression analysis of the TEs identified 343 significantly differentially expressed elements between the ecotypes (90 upregulated and 253 downregulated) (Figure 2, Appendix A). These included 76 LTR/Gypsy, 16 LTR/Copia, 9 LTR/unknown, 125 DNA transposons, and 117 unknown elements. Thirty of these differentially expressed TEs were located near or within genes (Appendix A). Overall, the TE expression levels were significantly lower than the gene expression levels (*t*-test, *p* = 1.91 × 10^−7^).

### 3.6. Transposable Element Insertion Polymorphisms Among the Two Ecotypes

Stringent filtering identified 3859 high-confidence TIPs, including 3615 non-reference TE insertions and 244 reference absences (Appendix A). Most of the TIPs were attributed to Gypsy (2764) and Copia (231) retrotransposons and the DNA transposon superfamilies DNA/DTM (488) and DNA/DTA (128) (Figure 3A). The site frequency spectrum (SFS) analysis showed 1.62% of the TIPs had a minor allele frequency < 0.05, though this may be underestimated due to the sample size (Figure 3B).

The TIP distribution analysis revealed significantly higher variant frequencies in intergenic regions (1699), compared to introns (415), UTRs (264), and CDSs (95). Across each category, the variant frequency exhibited a consistent decrease as the MAF bins increased from 0–0.1 to 0.4–0.5 (Figure 3C). Notably, specific superfamily groups exhibited similar patterns: 2764 TIPs in LTR/Gypsy, 488 in DNA/DTM, 231 in LTR/Copia, 156 in Helitron, 128 in DNA/DTA, 56 in DNA/DTC, and 28 in LTR/Unknown (TIP counts > 10 for each superfamily) (Figure 3D). Across all the superfamilies, a consistent negative correlation between the variant frequency and the MAF bins was observed, indicating a purifying selection on these TIPs (Figure 3D). The PCA of the TIPs demonstrated a significant genetic differentiation between the HD and YC populations, highlighting the role of the TIPs in distinguishing genetic ancestry (Figure 3E). The LD analysis with the nearby SNPs revealed that 51.4%, 39.0%, and 9.6% of the TE variants exhibited low, intermediate, and high LD, respectively (Figure 3F). A positive correlation was observed between the MAF and the LD, with the higher MAF variants more frequently exhibiting a high LD (Figure 3F). This pattern was consistent for both TE insertions and deletions (Figure 3G). Notably, the LD category assignments were largely consistent between insertions and deletions (Figure 3G), supporting the observation that common alleles tend to be in high-LD states.

Twenty-nine fixed TIPs associated with 28 genes were identified between the red ecotype and white ecotype populations (Appendix A). These TIPs were distributed as follows: 22 intergenic, 3 intronic, and 2 each in CDS and UTR regions. The affected genes included homologs of transcription factors (AP2/ERF, bHLH, MYB-like, and C2H2) and enzymes (Homocysteine S-methyltransferase (HMT), alpha/beta hydrolase fold, and 2OG-Fe (II) oxygenase superfamily) (Appendix A). Three genes with fixed TIPs were differentially expressed between the white and red color ecotypes. *Barthea13541* (log|fold change| = −2.70, *p*-adj = 0.0003), containing a non-reference DNA/DTM insertion in the white ecotype population, is homologous to the Pmr5/Cas1p GDSL/SGNH-like acyl-esterase family protein, influencing cell wall modification, plant–pathogen interactions, stress responses, and development [58]. *Barthea35147* (log|Fold Change| = 3.90, *p*-adj = 0.002), with a fixed DNA/DTM present in HD but absent in YC, is homologous to the AP2/ERF transcription factor, regulating plant morphogenesis, stress responses, hormone signaling, and metabolism [59,60]. *Barthea36291* (log|Fold Change| = 4.05, *p*-adj = 0.003), with a fixed DNA/DTA present in the white ecotype but absent in the red ecotype, is homologous to Cytochrome P450 (CYP), synthesizing compounds crucial for membrane structure, hormones, UV protection, pigments, signaling, and volatile compounds mediating biotic and abiotic interactions [61].

## 4. Discussion

### 4.1. Impact and Distribution of Transposable Elements in the Compact Genome of B. barthei

The *B. barthei* genome (235.03 Mb) exhibits a compact architecture with a low transposable element content, consistent with other streamlined plant genomes. Its TE proportion aligns with the broad spectrum observed across plant genomes, ranging from 2.5% in *U. gibba* [7] to 85% in maize [62]. Within this compact genome, the Ty1/Copia and Ty3/Gypsy superfamilies dominate, distinguished by divergent reverse transcriptase (RT) and integrase (INT) domain arrangements in their POL polyproteins. The predominance of Ty3/Gypsy elements across Viridiplantae [63] may reflect their distinct genomic dynamics: older Ty3/Gypsy elements typically localize heterochromatin, whereas younger Ty1/Copia elements preferentially occupy euchromatic regions with a reduced recombination suppression [64,65].

The Class II DNA transposons in the *B. barthei* genome encompass five major superfamilies but lack the P superfamily. Their abundance varies markedly among plant species [66], with TIR and Helitron transposons showing greater interspecific variability than LTR retrotransposons. For instance, Arabidopsis harbors more Helitron elements than TIR elements, and the DNA transposon content can diverge significantly even between closely related taxa [67]. Although DNA transposons typically constitute a small fraction of plant genomes, their frequent gene-proximal localization and mutagenic potential profoundly influence genome architecture [66]. Notably, Mutator-like elements (MULEs), which elevate mutation rates up to 50-fold in plants [68], dominate *B. barthei*’s DNA transposon repertoire. Pack-MULEs further drive biased gene modifications through targeted insertion and sequence capture [4], positioning these elements as key contributors to *B. barthei*’s genomic plasticity.

Most of the TEs in *B. barthei* reside in intergenic regions (Table 2), with merely 577 copies (0.9%) localized to promoters including putative cis-regulatory elements. While the majority appear linearly distal from genes, chromatin spatial organization (e.g., 3D folding) may mediate their physical proximity to transcriptional units [69]. Intragenic TEs (6569 copies, 10.3%) likely reshape regulatory networks by introducing cryptic splicing sites, facilitating exonization, or embedding novel regulatory motifs [70,71]. Collectively, TEs orchestrate gene expression through both sequence-encoded features and chromatin-mediated spatial interactions, forming a dynamic regulatory framework that amplifies transcriptional diversity.

Our genome-wide TE characterization reveals their dual evolutionary role in *B. barthei* as generators of adaptive variation and potential genomic destabilizers. The compact genome architecture may reflect an evolutionary equilibrium between TE-driven adaptability and stability maintenance. While this study delineates the TE landscape, the ecological relevance of these elements—particularly under environmental stressors or niche-specific pressures—remains an open question, warranting functional studies to dissect their contribution to adaptive evolution.

### 4.2. TE Islands Facilitate B. barthei Adaptation to Tropical Forests

Despite their sparse gene distribution, TE islands in *B. barthei* exhibit non-random gene enrichment, notably in photosynthesis, tryptophan metabolism, and abiotic stress tolerance pathways. Photosynthesis, which is essential for carbon fixation, is highly sensitive to abiotic stressors such as ultraviolet (UV) radiation, fluctuating light intensity, and oxygen deprivation [72]. Tryptophan metabolism plays a key defensive role against pathogens, as shown in rice and Arabidopsis [73,74]. Its enrichment in *B. barthei*’s TE islands may provide an evolutionary advantage against pathogens and herbivores, consistent with the Janzen–Connell hypothesis regarding tropical forest diversity [75].

*B. barthei*’s distribution, extending into the subtropical regions of southern China, aligns with MacArthur’s (1972) ecological paradigm, which posits that species’ ranges are limited by abiotic factors at one extreme and biotic pressures at the other [76]. *B. barthei* has evolved adaptations to overcome these constraints, including enhanced photosynthetic capacity, polymorphic flower colors, and diverse growth forms (personal observations). High-altitude populations, including the red-flowered ecotype from Huidong, Guangdong, exemplify these adaptions. Adapted to mountain mists, variable rainfall, strong winds, and intense sunlight, this ecotype displays deep red flowers, thick coriaceous leaves, and a compact growth structure.

While the adaptive significance of TE islands is documented in species such as the invasive ant *C. obscurior* [13], where they facilitate a relaxed selection and genetic modifications for enhanced chemical perception, learning, and insecticide resistance, their role in plant adaptation is less clear. In *B. barthei*, the observed flower color and leaf texture diversity suggests that the TE islands may be crucial for niche expansion. Further research is needed to understand how the TEs adjacent to candidate genes regulate these adaptive traits, revealing how TE islands contribute to *B. barthei* ‘s successful adaptation to tropical forest environments.

### 4.3. Transposable Element Insertion Preferences and Polymorphisms Associated with Ecological Divergence Between Ecotypes

The distribution of TEs in *B. barthei* reveals distinct insertion patterns and their potential role in ecological adaptation. While only 26.1% of *B. barthei* genes are associated with TEs, compared to 78% in maize [77], both species share similar TE distribution patterns, with most copies within 5 kb of genes. Our survey identified 3859 high-confidence TIPs, primarily Gypsy (2764) and Copia (231) retrotransposons and DNA transposon superfamilies DNA/DTM (488) and DNA/DTA (128). The TIP distribution mirrors TE insertion preferences, with most in intergenic regions (1699) followed by introns (415), UTRs (264), and CDS regions (95).

The non-random TE distribution reflects the interplay of natural selection and genetic drift [78]. The decreasing TIP variant frequency with increasing MAF suggests an ongoing selection, especially against deleterious gene body insertions. This is supported by the LD patterns: 51.4% of the TE variants showed a low LD with nearby SNPs, and only 9.6% exhibited a high LD. The positive MAF–LD correlation suggests that beneficial or neutral TE insertions are more likely to persist and become common variants.

The TE insertion impact varies by genomic location [79]. Regulatory region (promoter, enhancer) insertions can result in gene expression nullification, enhancement (via new cis-regulatory sites), or silencing (via repressive chromatin marks) [80]. Examples include a Copia-like element insertion upstream of the *ruby* gene, a key MYB transcriptional activator that affects anthocyanin production in blood oranges [81] and two independent insertions by a MITE and an LTR element at the *teosinte branched1* (*tb1*) locus altering maize ear morphology during domestication [82].

The significant genetic differentiation between the red ecotype and the white ecotype populations (TIP-based PCA) highlights the role of TE polymorphisms in ecological divergence. The 29 fixed TIPs associated with 28 genes, including three differentially expressed genes, may contribute to ecotype-specific adaptations. The fixed DNA/DTM insertion in *Barthea13541* (YC) potentially affects cell wall modification and stress responses. Similarly, the ecotype-specific insertions near *Barthea35147* (AP2/ERF homolog) and *Barthea36291* (Cytochrome P450 homolog) likely contribute to divergent morphogenesis, stress responses, and secondary metabolism between the white and red ecotypes.

While intron-nested insertions (8.6% of total) are linked to phenotypic variation (e.g., double flower in morning glories [83] and yellowhorns [84]), CDS insertions are rare due to a strong negative selection. The few surviving CDS-nested TEs, such as the Copia insertion affecting soybean salt sensitivity [85], are typically ancient and conserved. The insertion distribution, combined with fixed inter-ecotype polymorphisms, suggests that TEs have significantly contributed to *B. barthei* adaptive divergence across ecological niches.

## 5. Conclusions

Our comprehensive analysis of the *B.barthei* TE landscape reveals key insights into genome organization and ecological adaptation. The relatively compact *B. barthei* genome (235 Mb) has a low TE proportion with distinct distribution patterns dominated by Ty3/Gypsy retrotransposons and Mutator-like DNA transposons. Despite their limited abundance, these TEs exhibit a non-random distribution, with most within 5 kb of genes, suggesting their potential regulatory roles in genome evolution and adaptation.

The identification of TE islands enriched in photosynthesis, tryptophan metabolism, and stress response genes provides compelling evidence for their role in ecological adaptation. These genomic features likely facilitated *B. barthei*’s successful colonization of diverse tropical and subtropical habitats in southern China, as evidenced by distinct ecotypes including the red-flowered variant in Huidong.

An analysis of 3859 high-confidence TIPs revealed a significant genetic differentiation between between the HD and YC ecotypes. The presence of twenty-nine fixed TIPs in 28 genes (involved in cell wall modification, stress responses, and secondary metabolism) suggest that TE-mediated genetic variation has substantially contributed to the ecological divergence of *B. barthei*. The observed TE distribution and polymorphism patterns (preferential intergenic insertion, strong selection against CDS insertions) reflect the balance between adaptive variation and genome stability.

These findings advance our understanding of the contribution of TEs to plant genome evolution and adaptation, particularly in tropical forests. Future research should investigate the specific regulatory mechanisms by which TE insertions influence adaptive traits, potentially revealing the role of mobile genetic elements in plant speciation and ecological divergence.

## Figures and Tables

**Figure 1 biomolecules-15-00346-f001:**
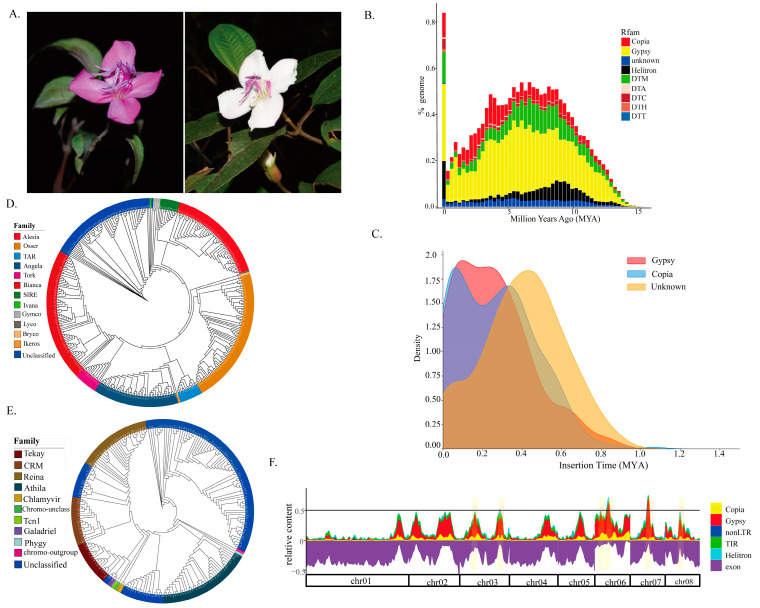
Transposable element landscape of *Barthea barthei*. (**A**) Photographs of the two ecotypes of *B. barthei*: white ecotype (**right**) and red ecotype (**left**); (**B**) the age distributions of different superfamilies measured by divergence with corresponding consensus sequences, including superfamily Copia (red), Gypsy (yellow), DTM (green), Helitron (black), DTA (light plink), DTC (dark red), DTH (brown), DTT (light blue), and Unknown (blue); (**C**) insertion time distribution for intact LTR retrotransposons, including superfamily Copia (blue), Gypsy (red), and Unknown (unclassified LTR, yellow); (**D**) classifications for LTR retrotransposons based on phylogenies of reverse transcriptase (RT) domains for superfamily Ty1/Copia, including family *Alesia* (red), *Osser* (orange), *TAR* (light blue), *Angela* (dark blue), *Tork* (magenta), *Bianca* (bright red), *SIRE* (green), *Ivana* (light green), *Gymco* (gray), *Lyco* (dark gray), *Bryco* (light orange), *Ikeros* (dark orange), and Unclassified (blue); (**E**) classification for superfamily Ty3/Gypsy, including family *Tekay* (dark red), *CRM* (brown), *Reina* (dark brown), *Athila* (teal), *Chlamyvir* (olive green), *Chromo-unclass* (light green), *Tcn1* (bright green), *Galadriel* (purple), *Phyggy* (light blue), *chromo-outgroup* (pink), and Unclassified (blue); (**F**) relative content of exonic and TE-derived sequences along the eight largest scaffolds of the *B. barthei* genome. Shown are DNA transposons (DNA), long interspersed nuclear element (LINE) and LTR retrotransposons, as well as other TEs (other). The genome is well structured into TE-poor regions (’low-density regions’, LDRs), TE-rich regions (’TE islands’, orange highlights), and genome features such as Copia (yellow), Gypsy (red), non-LTR (blue), TIR (green), Helitron (cyan), and exon (purple).

**Figure 2 biomolecules-15-00346-f002:**
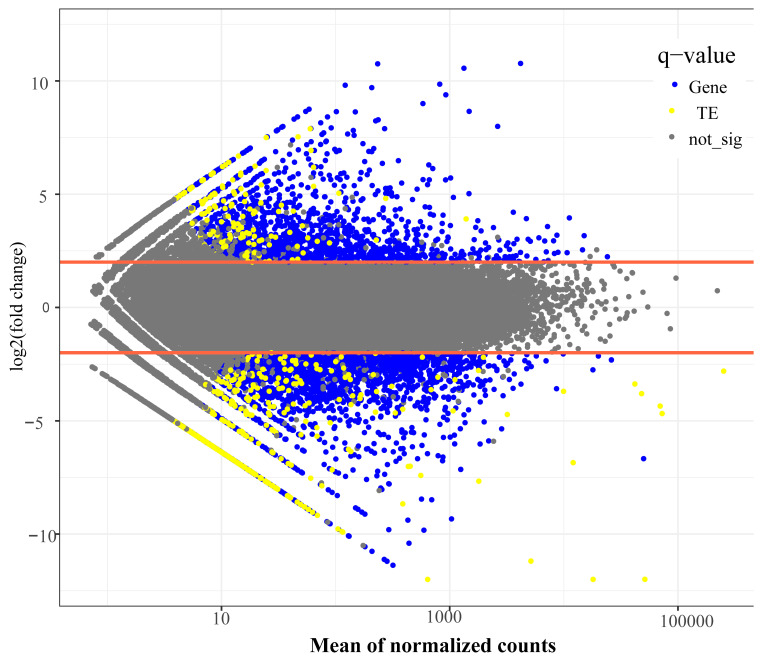
The distribution of differentially expressed genes and transposable elements between the white ecotype and red ecotype during flower bud development of *Barthea barthei*. The significant levels were determined using a |log2FoldChange| > 2 (as delimited by the red lines) and an adjusted *p*-value of 0.05.

**Figure 3 biomolecules-15-00346-f003:**
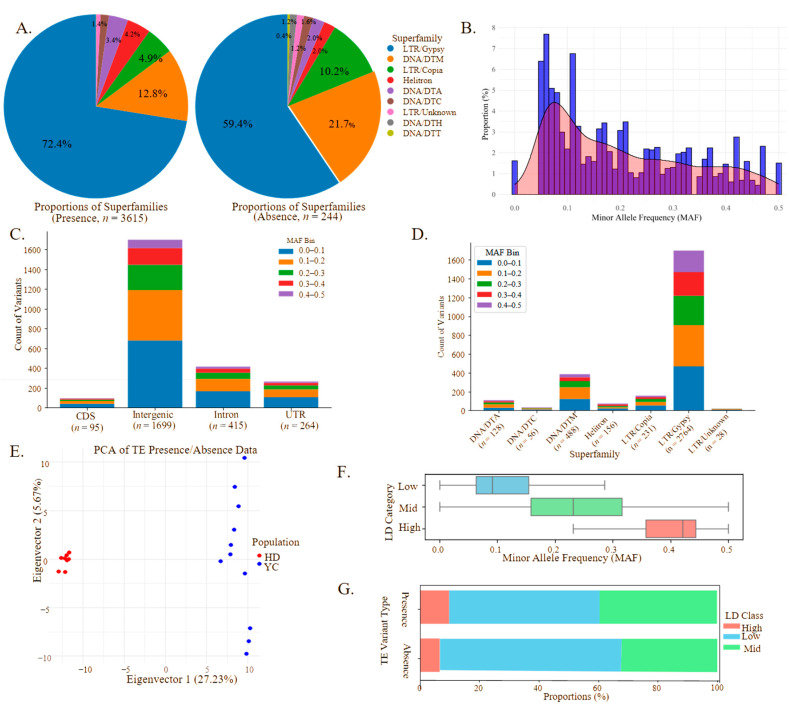
Transposable element insertion polymorphism patterns between the white ecotype from the YC population and the red ecotype from the HD population of *Barthea barthei*. (**A**) Superfamily components and proportions of non-reference transposon element insertion variants (**Left**) and reference absence variants (**Right**). The different superfamily color codes are as follows: LTR/Gypsy (dark blue), LTR/Copia (green), LTR/Unknown (pink), DNA/DTM (orange), DNA/DTA (purple), DNA/DTC (brown), DNA/DTH (gray), DNA/DTT (olive green), and Helitron (red). (**B**) The minor allele frequency (MAF) distribution of transposable element insertion polymorphisms for *B. barthei.* (**C**) Counts of TE variants with different minor allele frequencies within each genomic feature classified as coding regions (CDS), intergenic regions, introns, and untranslated regions (UTRs). Color coding for MAF bins are as follows: 0.0–0.1 (blue), 0.1–0.2 (orange), 0.2–0.3 (green), 0.3–0.4 (red), and 0.4–0.5 (purple). (**D**) Counts of TE variants with different minor allele frequencies within each TE superfamily including LTR/Gyspy, LTR/Copia, LTR/Unknown, DNA/DTM, DNA/DTA, DNA/DTC, and Helitron. Color coding for MAF bins is as in Panel C. (**E**) Principal component analysis for the samples of red ecotype (HD population, red dots) and white ecotype (YC population, blue dots) based on non-reference TE insertion variants and reference TE absence variants. (**F**) Minor allele frequency distribution by relative TE–SNP linkage disequilibrium. Boxplots of minor allele frequency (MAF) for genetic variants are grouped by the relative linkage disequilibrium (LD) of nearby transposable elements (TEs) with SNPs. LD categories (Low, Mid, High) are based on the ranking of TIP–SNP r^2^ values relative to the median ranked SNP–SNP r^2^ within the same region. Boxes show interquartile range (IQR), with median indicated; whiskers extend to 1.5× IQR. (**G**) TE variant LD class distribution by presence/absence. Stacked bar chart showing the proportion of transposable element (TE) variants (TIPs) in each relative linkage disequilibrium (LD) class (High, Low, Mid) for two TIP states: Presence and Absence. LD classes are defined relative to regional SNP–SNP LD. Bars represent 100% for each TIP state.

**Table 1 biomolecules-15-00346-t001:** Summaries of repeat contents in the genome *Barthea barthei*. Intact transposable elements are shown in the parentheses.

Class	Order	Superfamily	Number	Total Length (bp)	Percentage of the Genome (%)
Retrotransposon	LTR				
		Copia	9602 (120)	5,809,421 (665,156)	2.47 (0.28)
		Gypsy	24,293 (220)	20,729,316 (2,036,809)	8.82 (0.87)
	DIRs				
		DIRs	1	254	0
	LINE				
		L1	588	277,520	0.12
		I	8	13,274	0.01
	PLE				
		Penelope	278 (1)	176,212 (4083)	0.07 (0.00)
	SINE				
		tRNA	220	21,226	0.01
DNA Transposon					
	TIRs				
		hAT	516 (56)	222,301 (98,002)	0.09 (0.04)
		CACTA	2820 (112)	1,144,474 (310,909)	0.49 (0.13)
		PIF_Harbinger	747 (30)	282,584 (92,465)	0.12 (0.04)
		Mutator	13,868 (296)	6,337,070 (729,432)	2.7 (0.31)
		Tc1-Mariner	127 (14)	97,949 (41,156)	0.04 (0.02)
	MITEs				
		DTA	455 (25)	107,235 (8582)	0.05 (0.00)
		DTC	7 (6)	2317 (2196)	0 (0.00)
		DTH	70 (8)	11,773 (2513)	0.01 (0.00)
		DTM	1769 (130)	360,581	0.15
		DTT	1 (1)	221 (221)	0 (0.00)
	Helitron				
		Helitron	9494 (89)	3,834,769 (984,799)	1.63 (0.42)
Other_repeats					
	Other_repeats	Other_repeats	19,316	4,942,020	2.1
Total					18.88

**Table 2 biomolecules-15-00346-t002:** The summaries of coordinate relationships between different orders or superfamilies of transposon elements and the nearest host genes in the genome *Barthea barthei*.

		Copia	Gypsy	TIR	Helitron	nLTR	Total	Proportion (%)
Upstream							29,144	45.8
	Promoter	64	95	254	161	3	577	0.9
	other	4480	11,422	8599	3970	96	28,567	44.9
Downstream		4185	10,611	7999	3635	132	26,562	41.7
5′-Overlap		71	246	281	159	3	760	1.2
3′-Overlap		61	249	209	111	0	630	1.0
Nested							6569	10.3
	Single intron	543	1333	2347	1213	18	5454	8.6
	Single 5′ UTR	43	47	59	41	0	190	0.3
	Single 3′ UTR	61	112	119	78	0	370	0.6
	Single CDS	3	23	37	12	0	75	0.1
	Exon/intron	91	155	123	108	3	480	0.7
Total		9602	24,293	20,027	9488	255	63,665	

**Table 3 biomolecules-15-00346-t003:** Coordinated expression of transposable element transcripts and nearby/nested genes during flower bud development in *Barthea barthei*.

Class	Order	Superfamily	Copy Number	Average BaseMean	^a^ Instances of Coordination
Class I (Retransposons)					
	LTR		1457	9.85	
		Copia	206	5.62	69:7:2:54:2:7:1:64
		Gypsy	1059	10.55	392:18:4:207:13:0:4:24
		Unknown	148	9.37	33:4:0:55:2:0:0:4
	LINE	Unknown	44	14.53	4:1:0:30:3:0:0:6
Class II (DNA transposons)					
	TIR		1149	16.67	
		DTA	45	13.1	11:5:0:19:0:0:1:9
		DTC	403	18.43	57:30:0:235:7:2:16:56
		DTH	48	13.14	8:10:0:17:0:0:2:11
		DTM	626	8.43	174:25:4:216:3:0:38:165
		DTT	27	17.67	9:2:0:2:0:0:3:11
	MITE		140	6.17	
		DTA	68	8.15	5:0:0:58:1:0:0:4
		DTC	3	2.1	0:0:0:1:0:0:0:2
		DTH	13	4.51	2:1:0:8:0:0:0:2
		DTM	56	4.37	22:2:0:18:0:0:2:12
		DTT	0	0	
	Helitron		649	24	136:47:2:264:8:3:39:150
Unknown			1083	573	289:70:7:360:31:6:34:316
pararetrovirus			17	5.24	4:0:0:5:1:0:0:7

^a^ Upstream:5′-overlap:5′UTR: intron:intron–CDS (CDS):3′UTR:3′-overlap:downstream.

## Data Availability

The raw data of whole genome sequencing and RNA sequencing are deposited in the Genome Sequence Archive (GSA accession: CRA012896 under the project PRJCA020264) in the China National Genomics Data Center (NGDC) database. The genome assemblies and annotations are available at [https://github.com/altingia/Barthea_TE_manuscript accessed on 23 January 2023] under accession number GWHDUDN00000000.

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
