# Peer review of "Transposable Element Landscape in the Monotypic Species Barthea barthei (Hance) Krass (Melastomataceae) and Its Role in Ecological Adaptation"

_biomolecules, 2025, doi:10.3390/biom15030346_

Round 1
Reviewer 1 Report
Comments and Suggestions for Authors
Wu and colleagues reported in this manuscript their interesting results from profiling the transposable elements (TEs) in two ecotypes of Barthea barthei. Specifically, they performed the identification, classification of TEs, and expression of TEs along with that of genes in the genomes of the two ecotypes. They also examined polymorphic TE insertions in population samples. The research design is sound, and the data analysis is comprehensive, providing interesting new data in supporting the role of TEs in plants’ adaptation to environment. The manuscript is well written in general, but suffer a large number of typographical, grammatical, and clarity issues. I also think the discussion section could be improved by adding the coverage of a few points as suggested below.
Critical/scientific issues
- Referencing is missing or insufficient in various places, e.g., for the statement in L59, L62-64, …; for the ORF finder program (L150).
- L178-179: Specific criteria for defining the categories, especially for 5-overlap, 3-overlap, and downstream (could cite Figure S2).
- In section 2.1, the number of samples for each experiment needs to be provided here.
- L203-204: please clarify whether a read is used only once to go with the best match and how to handle those with multiple equal matches as the best match.
- Figure 1E: I can’t seem to understand why copia showed both positive and negative value for the same locations, e.g., chr03, 06-08.
- In section 3.6, it should be useful to add (perhaps as an additional panel in Figure 3) the rate of TIPs (in the total copies of the family) for each of TE families to measure and compare the relative ongoing transposition activity.
- L361: some discussions could be added regarding the implication of “TE copies spanning multiple introns/exons”, e.g., whether these exons were contributed by the TEs.
- How to interpretate the coordinated expression of genes and nearby TEs? Can we say that TE expression enhanced the expression of the gene or the other way around?
- LTRs often display trimorphism (presence and absence, plus solo-LTR vs full-length). Is there any information regarding the length of LTR TIP variants as an indication of possible trimorphism?
- Are TIP variations as TE insertions in CDS mostly predicted to have frameshift impact on the host genes? Could any of these explain the loss of color in the white ecotype? This lack of citing specific data in supporting statements specific to B. barthei is seen in a few other sections in the discussion, e.g., the 3rd last and last paragraph in 4.3
- Cite the specific data supporting the statement in L502-503. Same is true for a few other places in the discussion section.
- L611: regulatory mechanism is not the only way TE insertion impact gene function. It might be easier and at least interesting to find out the contribution of genes with TEs in CDS to the creation of the ecotypes.
Typographical and grammatical issues:
- Issues with use of acronyms including lack of defining before using: L49: “DIR”; repeated defining: “LTR” (L75) was already defined in L45, “TE” (L79, 117, 324) was already defined very early on; or improper uses: “TE elements” should be “TEs” (L117), Penelope (L256).
- Extra space after most ref citation numbers (L45,48,60….), unnecessary comma after “type” in L46.
- Inconsistencies in numbers: some with comma for thousand, and some not; the former should be preferred.
- L91, an “and” is needed before “chromatin”.
- There is no need to capitalize and hyphenize “liquid nitrogen”.
- L133: better replacing “150bp” with “150bp x 2”.
- L139, 244, 287, 315: “B. barthei” needs to be italicized (my list may not be complete)
- L146: “a significance E-value of 0.001” should be better written as “a minimum of E-value of 0.001”.
- L173-174: “only those….” should start as a new sentence.
- L174: “less upstream 1000 bp to” is awkward and can be changed to “less than 1000 bp upstream to”
- L175: “insertion” should be used in plural form to match the subject.
- L286 & 289-290: an extra space in “TE- derived” and “low- density”.
- L294: grammatical error in “to constructed”.
- L334: “The study identified.” as extra?
- L354: extra “’” after “beta”.
- L356: no need to capitalize the first letters.
- The titles for some supplementary tables have irregular capitalization (e.g., Enriched); considered changing “genome assembly Barthea barthei” to “the genome assembly of Barthea barthei (for multiple tables).
- L476: “dominate” should be a better word in place of “predominate”.
- L512: “or” needs to be replaced with “and” to go with the prior “both”.
- Table 1 & 2: splitting across pages can be avoided.
- Table S20: the chromosomal/contig ID is mission from the entries before the start & end coordinates.
- Figure 1: the letter labels for subpanels are missing in the figure. The caption for the color scheme is missing for B & D; readers need to be able to link the group IDs described in section 3.2 to the clusters in panel D.
- Figure 3: The format/alignment of the legend text is messed up; “below” at the end of L437 better written as “the bottom subpanel” to be clear.
- Figure S1 text legend: ”continued to the main text” should be “continued from Figure 1E” instead.
- Figure S2 text legend: “Schematic of” should be “Schematic representation of”; the meaning of 5’G and 3’G was not explained. Assuming it’s for 5’ and 3’ end of the gene, respectively, then Upstream should have 5’G.
- Figure S3: in the figure legend title, “their flanking nearest” better written as “their nearest flanking”.
- Figure 4S: the case of the letter labels in the figure (upper case) doesn’t match that in the legend text (lower case).
(see lists above)
Author Response
pls see the attachment

Reviewer 2 Report
Comments and Suggestions for Authors
This study investigates transposable element (TE) insertions and the possible role of TE insertions in environmental adaptation by the Barthea barthei plants. The authors used previously published genomic sequence data plus newly generated RNA-seq data to identify TEs, characterize TE insertions, compare the expression of genes and TEs between two different ecotypes of Barthea barthei. They show that genes associated in TE-rich regions of the genome (TE islands) are enriched for plant defence categories. They also show that the two ecotypes have differential gene and TE expression and display significant TE insertion polymorphisms. Based on their results they conclude that TE insertions play a role in genome evolution and ecological adaptation of Barthea barthei under the tropical forest environments. While the paper has not provided specific connection between TE insertions and the distinct phenotypes of the two ecotypes, which is a weakness, the good-quality data on TE classification, TE insertion, gene expression and TE polymorphisms does provide a good foundation for further studies of such specific links. The paper is in general well written with good quality of English.
Specific comments:
- 1st paragraph of the Introduction can be supported with a simple diagram summarizing the hierarchy of TE families and subfamilies described in the text, making it easier to follow.
- “Red flower ecotype” and “purple ecotype” are both used to describe the same plant genotype. Only one should be used for consistency.
- Line 452: what does “fixed TIPs” mean? Please provide a brief explanation.
- Line 52: Change “According the” to “According to the”
- Line 73-74: The following sentence is a little hard to understand: “Despite having low transposable element diversity at the family level, the Norway spruce genome was dominated by…..”. May delete the first part “Despite having low transposable element diversity at the family level,”
- Lines 104-105: Change it to “specifically in the provinces of Hunan, Guangdong, Guangxi, Fujian, and Taiwan” to avoid repeated use of “Province”.
- Section 2.1: “White petal ecotype” is not mentioned, so there must be a typing error.
- Line 150: Change “over 50” to “over 50 amino acids”?
- Line 203: Change “For both TEs and genes” to “Both TEs and genes”
- Figure 1 is not labelled with A, B, C, etc, which should be corrected.
- Line 294: Change “to constructed” to “to construct”
- Line 334: Delete “The study identified.”
- Line 407: Change “Differential express analysis of TEs” to “Differential expression analysis of TEs”
- From line 438 on: It is better to use “white ecotype” and “purple ecotype” rather than “YC population” and “HD population” in the text.
Author Response
pls see the attachment.

Reviewer 3 Report
Comments and Suggestions for Authors
Dear Authors,
Thank you for submitting your manuscript. In the present work, you investigate the transposable element (TE) landscape in the non-model plant B. barthei by leveraging a series of bioinformatics tools and analytical pipelines. You begin by quantifying the TE species, copy numbers, and their ages in B. barthei. Additionally, 13 TE hotspots with over 50% TE coverage were identified, which were found to enrich genes involved in photosynthesis and the tryptophan metabolism pathway. Next, the genomic distribution of TE insertions was surveyed, revealing distinct insertion patterns. To investigate the driving factors between ecotypes, RNA-seq analysis was conducted, which identified 2,544 differentially expressed genes (DEGs) and 343 TEs. The study also explored the TE insertion polymorphisms (TIPs) between the ecotypes and the affected genes.
By analyzing the TE landscape, the genes enriched in TE islands, and the TIPs across ecotypes, the authors demonstrate that, TE, and TE-mediated genetic variations, play a significant role in the ecological adaptation of B. barthei.
Overall, the manuscript is well-written, and the methods are adequately documented. I have the following specific suggestions:
- Please proofread the manuscript carefully to correct typos, grammatical issues, inconsistent styles, etc. Below are a few examples:
- Line 51, Line 174, Line 372, Table S18/S19/S22 title (grammar issue).
- There seems to be extra space after most citations.
- Line 426-427: formatting issue.
- "Table 1" and "Table 3" should be in bold (inconsistent styling).
- Some occurrences of B. barthei are not italicized.
- Line 155: "Customer scripts" should be "Custom scripts"; "Github" should be "GitHub".
- In Table 3: "Line" should be "LINE".
- Figure 1 lacks legends and sub-plot labels. In Figure 3A, the labels are overlapping with each other.
- Please provide a figure that illustrates the major steps, their goals, and the corresponding analytical pipelines used in this work. This would help readers gain a high-level overview of the research.
- The TE insertion landscape in the B. barthei genome is surveyed, but the statistical analysis of whether TEs are enriched in specific genomic elements is missing.
Author Response
pls see the attachment

Reviewer 4 Report
Comments and Suggestions for Authors
The manuscript investigates the role of TEs in Barthea barthei using genomic, transcriptomic, and population genomics approaches. It highlights the significant proportion of the genome occupied by TEs and their role in adaptive traits, particularly the ecological adaptation between red and white flower ecotypes. However, several issues related to consistency, clarity, and methodology need to be addressed.
1. Improper useage of terminology, such as transposable elements (TEs) are introduced at the start of the manuscript. After this initial mention, the abbreviation 'TEs' should be used consistently throughout the manuscript, rather than alternating between 'Transposable elements,' 'TEs,' or 'Transposable elements (TEs)' at different points in the text. Similar issue for TE insertion polymorphisms (TIPs), million years ago (MYA), LTR (long terminal repeats), minor allele frequency (MAF), linkage disequilibrium (LD).
2. LINE 69, "ranging from less 3%......to over 70% of the giant genome of Norway spruce (Picea abies, 19.6 Gb)", shouldn't use maize (85%) as maximum percentage of TE content
3. The introduction requires a rephrasing to enhance clarity and flow. The first section outlines the classification of TEs, followed by a discussion on the TE content and their age distribution within genomes. The next part highlights how TEs contribute to trait variation, and the final paragraph introduces the study species, Barthea barthei, its geographic distribution, and population structure. However, the manuscript’s purpose is not clearly articulated. Prior to the methodology section (LINE 115), the authors should refine and clarify the objective of the study, providing a clearer context. Additionally, they should identify key research gaps and unresolved questions to guide the reader through the significance of the research.
4. Discrepancies in TE Counts:
LINE 250, "Further analysis with DeepTE successfully assigned 5665 of these LTR/unknown copies to Ty3/Gypsy (3768 copies) and the Ty1/Copia (1897 copies) superfamilies (Table 1)." it is not mentioned in the table. What's more, it is very confusing that in the final Table 1, the Copia count is reported as 9602 and the Gypsy count as 24293. This raises a concern, as the reported Gypsy count (24293) minus the 3768 assigned by DeepTE exceeds the total number of LTRs (5923) initially identified by EDTA. How is this discrepancy accounted for?
LINE 265, the manuscript reports "These including 120 LTR/Copia, 220 LTR/Gypsy, 1 nonLTR/PLE, 426 TIR/DTM, 118 TIR/DTC, 81 TIR/DTA, 38 TIR/DTH, 15 TIR/DTT, and 89 Helitron." However, this contradicts the later statement in LINE 273, "We identified 782 intact LTR elements, consisting of 188 Ty1/Copia, 381 Ty3/Gypsy, and 213 unclassified elements. " This raises a concern: how could the number of intact LTR elements exceed the total reported in line 265, as well as the counts in Table 1? This discrepancy needs clarification.
5. Figure 1 issues. There is no clear annotation for the panels (A, B, C, D, E), and the figure lacks an explanation for several key aspects. For Figure 1B (assuming this is the correct reference), the x and y axes are not labeled, and color coding used in the visualizations is unclear. A similar issue arises in plots, where the significance of the various colors is not explained, and the label 'Unknown' is used without clarification. Additionally, for the intact LTR retrotransposons, the manuscript does not provide a total count, which should be included for better understanding.
6. Methodological Clarification: LINE 304, TE islands detection is based on " we analyzed the genome using 500 kb sliding windows with a step size of 100kb along each linkage group." However, it would be beneficial to include a permutation test to assess the significance of these TE islands. This test could help determine whether the observed patterns are statistically significant or if they arise by chance.
7. LINE 367, "Transcriptome analysis identified 40,538 expressed genes and 4495 transcribed TE copies". Does the number of 4495 transcribed TE copies refer to distinct loci? Assuming this is the case, as suggested by the explanation in LINE 388, where the authors describe the expression levels for different TE subfamilies "retrotransposons were predominant, with 1457 transcripts identified, including 1059 LTR/Gypsy, 206 LTR/Copia, 148 LTR/unknown elements, and 44 LINEs/unknown elements." Therefore, the KEGG pathway section seems somewhat disconnected. This part should either be rephrased or moved to a later section to ensure a more logical and smooth flow.
8. Figure 3 lacks a detailed legend and explanations for each plot. The manuscript should include clear descriptions of what each panel represents.
9. Formatting Issues: There is a consistent issue with extra spaces following each sentence, either before the commas or before the period.
10. TE Percentage in relative species: LINE 472, it is stated that TEs account for 16.76% (235,034,099 bp) of B. barthei's genome. Is this expected? It would be helpful to provide context by comparing this TE percentage with that of closely related species to better assess whether this proportion is typical or unusual for the genus.
